# The Role of Continuing Professional Training or Development in Maintaining Current Employment: A Systematic Review

**DOI:** 10.3390/healthcare11212900

**Published:** 2023-11-03

**Authors:** Rahman Shiri, Ashraf El-Metwally, Mikael Sallinen, Marjaana Pöyry, Mikko Härmä, Salla Toppinen-Tanner

**Affiliations:** 1Finnish Institute of Occupational Health, 00032 Helsinki, Finland; mikael.sallinen@ttl.fi (M.S.); marjaana.poyry@ttl.fi (M.P.); mikko.harma@ttl.fi (M.H.); salla.toppinen-tanner@ttl.fi (S.T.-T.); 2College of Public Health and Health Informatics, King Saud Bin Abdulaziz University for Health Sciences, Riyadh 14611, Saudi Arabia; ashraf.elmetwally@gmail.com; 3The Health Sciences Unit, Faculty of Social Sciences, Tampere University, 33720 Tampere, Finland

**Keywords:** education, employment, on-the-job training, personnel turnover, return to work, work engagement

## Abstract

The impact of continuing job education and professional development on early exit from the labor market is unclear. This systematic review examined how continuing job education or professional development influences the retention of current employment. We searched the PubMed and Embase databases from their start dates to January 2023. Two reviewers screened the full texts of relevant reports and assessed the methodological quality of the included studies using the adapted Effective Public Health Practice Project quality assessment. We qualitatively synthesized the results of the included studies. We screened 7338 publications and included 27 studies consisting of four cohort and 23 cross-sectional studies in the review. The participants of the selected studies were mostly from the health sector (24 studies). There were 19 studies on staying or leaving a current job, six on employee turnover intention, two on job change, one on return to work, one on early retirement, and one on employment. Continuing employee development or training opportunities were associated with increased intention to stay in a current job, decreased intention to leave a current job, decreased employee turnover intention, job change, or early retirement and with faster return to work. One of the two studies that examined the role of age showed that continuing employee development is a more important factor for retaining current employment among younger than older employees. A few studies found that job satisfaction and commitment fully mediated the relationship between employee development and employee intention to leave current employment. This study suggests that participating in professional training/development is related to a lower risk of leaving current employment.

## 1. Introduction

Employee exit from the labor market is influenced by various factors, such as personal, work, and organizational factors [1]. Age is a key factor that affects the risk of disability retirement, which is higher among older workers [1,2], and the intention to quit the current job, which is lower among older workers [3]. Education level also plays a role, as workers with lower education are more likely to leave the labor force due to disability, unemployment, or early retirement [1,4], while workers with higher education are more likely to change their current job [3,5]. Work-related factors, such as workload, working conditions, work–life balance, and burnout, affect employees’ intention to leave their job [3,6,7]. Psychological and organizational factors, especially low job control, are associated with disability retirement [8]. On the other hand, interventions such as adjusted job demands, social support at work, coaching, and job training can reduce the rate of premature exit from the labor market in workers with a chronic disease [1].

To keep and enhance their professional competence (knowledge and skills), workers need to engage in continuing professional development. This also helps them advance their careers, practice safely, provide better services to clients, and maintain consumer trust [9,10,11]. Continuing professional development is more common among health care workers [9], while its benefits for other occupations are less explored. Health care workers participate in continuing professional education and training to develop their careers, stay updated, and improve the quality of patient care [7,12].

Continuing professional development covers various short courses, conferences, workshops, seminars, and other short training programs. It can have different impacts on health professionals, such as increasing clinical knowledge; fostering networking and collaboration; changing attitudes; enhancing skills, competence, and performance; and influencing clinical practice [9,13]. By taking part in continuing education and training at work, workers can improve and refresh their skills and learn new ones [14]. On-the-job vocational training improved the mental health, sense of coherence, psychological stress, dysfunctional attitudes, and smoking rate among health care workers [15]. Employees who receive continuing job education or training report higher job satisfaction [16,17,18,19].

However, the effect of professional development and job education or training on staying or leaving the current employment is unclear. The purpose of this systematic review was to investigate how professional development and job education or training are related to maintaining or exiting the current employment. We also examined whether the relationship varies between younger and older workers.

## 2. Methods

### 2.1. Search Strategy

We followed the PRISMA guidelines [20] to design the review protocol. We searched PubMed and Embase from their start dates to 2 January 2023 using a combination of MeSH terms (PubMed), Emtree terms (Embase), and text words (Table 1). We also performed an extra search in Google Scholar. We did not apply any filters on the participants’ age or sex or on the publications’ language. We manually checked the references of the relevant articles on this topic to find more reports that might be useful.

### 2.2. Inclusion and Exclusion Criteria

Using PubMed, Embase, and Google Scholar, the first author searched for reports related to the topic and selected the ones that seemed relevant for further evaluation. Then, two reviewers (R.S. and A.L.-M.) screened the abstracts and full texts of the selected reports independently. The inclusion criteria were studies that investigated the effects of education, training, or job development on work retention or exit from paid employment using randomized or non-randomized controlled trials, cross-sectional, case control, and cohort designs. The exclusion criteria were studies that focused on vocational rehabilitation, employment services, and educational services for job seekers or people with a disability, as these interventions aimed at changing jobs rather than enhancing skills for the current job. Moreover, studies that used an organization as a unit of analysis and reported an employee turnover rate at the organizational level were excluded. Additionally, qualitative studies were not included in the review. The reviewers discussed any disagreements and reached a consensus.

### 2.3. Quality Assessment

The quality of the studies included in this review was evaluated by two independent reviewers (R.S. and A.L.-M.) using an adapted version of the Effective Public Health Practice Project quality assessment tool [21]. This tool assessed five types of bias: selection bias, performance bias, detection bias, attrition bias, and confounding (see Appendix A). The reviewers discussed and resolved any disagreements about the quality ratings.

### 2.4. Data Synthesis

We extracted the following characteristics from the studies that met the inclusion criteria for the review: study design, publication year, country of origin, study population description, sample age and sex distribution, sample size, professional education or training type, work participation or exit from paid employment, summary results, and confounding factors adjustment. We performed a qualitative synthesis of the results of the included studies because of the heterogeneity in professional training and outcome.

## 3. Results

A total of 3908 publications were retrieved from PubMed, and 3009 were retrieved from Embase (Figure 1). The first reviewer removed 579 duplicates and screened 6338 titles and abstracts from PubMed, Embase, and the first 1000 hits from Google Scholar (total: 7338). Google Scholar only allows screening the first 1000 results. Then, two reviewers assessed 145 abstracts or full-text articles for relevance. Out of those, 77 reports were excluded for not meeting the eligibility criteria, and 39 reports on vocational re-education or rehabilitation among job seekers or people with a disability and two studies on employee turnover rate were omitted from the review because they only reported the outcome at the organizational level and not at the individual level. Finally, the review included 27 studies consisting of four cohort studies and 23 cross-sectional studies. The participants of the selected studies were diverse but mostly from the health sector. Out of the 27 studies, 22 involved health care workers as the target population, while 2 focused on faculty members of health or medical sciences. The remaining three studies included people with a chronic disease and bank staff as participants.

The studies were published in different time periods. Eight studies were published between 2001 and 2010, eight were published between 2011 and 2015, and 11 were published between 2016 and 2022. The studies were conducted in various countries. Australia [16,22,23] and China [5,24,25] had three studies each. New Zealand [6,26], the United Kingdom [7,12], and the United States [27,28] had two studies each. Canada [29], Denmark [30], Ethiopia [31], Finland [32], Ghana [33], Japan [34], Italy [35], Pakistan [36], South Korea [37], Sweden [38], Taiwan [39], and the Netherlands [1] had one study each. One study recruited participants from Singapore and the USA [40], one study recruited participants from eight European countries (Belgium, Finland, France, Germany, Italy, Poland, Slovakia, and the Netherlands) [41], and another study was conducted in seven sub-Saharan African countries (Ethiopia, Kenya, Nigeria, Rwanda, Tanzania, Uganda, and Zambia) [42]. The number of participants in the included studies varied from 81 to 88,948.

The effects of professional development or training on job retention or turnover were assessed in three studies [27,28,30] using administrative data and 24 studies using self-reported data (Table A1 and Appendix A). The risk of selection bias was low in six studies, moderate in 11 studies, and high in 10 studies (Appendix A). Eighteen studies adjusted for some or all confounding factors. Attrition bias was low in all studies except two.

### 3.1. Job Retention

Five studies investigated the relationship between professional development or training and staying at the current job or intending to do so. An eight-year cohort study [28] reported that junior faculty members who participated in a development program were 11% more likely to remain at the same job than non-participants (67% vs. 56%, *p* = 0.04). Additionally, cross-sectional studies showed that professional development opportunities were linked to a higher intention to stay at the current job [16,22,29,42]. Professional development or a training opportunity was the main motivator for staying at a current job, and 80% of laboratory professionals from seven sub-Saharan African countries rated it as the most important or a very important factor for job retention [42]. Younger employees valued continuing professional development more than older employees for staying at the current job [22].

### 3.2. Leaving a Job

Four studies examined the association between professional development or training and leaving a job or the workforce, and 10 studies examined the intention to leave a job (Table A1). A large cohort study [27] found that women who attended 4-day early- and mid-career faculty professional development programs were less likely to leave their job than women who did not attend the programs. The programs had a positive effect on women’s job retention, as those who participated more than once were less likely to quit than those who only joined once. A similar finding was reported in a cohort study [12] in which employees who left their first job within six months cited a lack of study days (40%) and other courses (43%) as important factors in their decision. A cross-sectional study found that patients with rheumatoid arthritis who received additional job training after their diagnosis were less likely to leave the workforce than those who did not (adjusted OR 0.5, 95% CI 0.4–0.8) [1]. Dissatisfaction with development opportunities was also a major reason for nurses to leave their job in 51.4% of those who had left their institution [41].

Furthermore, a lack of professional development opportunities [31,32,33] and low perceived investment in employee development [39,40] increased the intention to leave a job. Some of the factors that influenced this intention were a lack of access to professional development [23,38], a lack of study opportunities [23], and a lack of access to courses other than study days [12] and/or study days [12]. A lack of professional opportunities ranked second after a low salary as a reason for leaving nursing care, and this was consistent between nurses aged < 45 years and those aged ≥ 45 years [38]. However, a cross-sectional study showed that a lack of career advancement and mandatory continuing professional development did not affect the intention to leave the dental nursing profession [7]. Another study also found no direct or indirect association between professional development and intention to leave an organization or profession through burnout and work engagement [6].

The relationship between the perceived investment in employee development and employee intention to a leave job was mediated by different factors in two studies [39,40]. Job satisfaction and affective commitment fully explained the relationship between perceived investment in employee development and the intention to leave a job for nurses [40]. For health care professionals in underserviced areas with a government subsidy program, the relationship between perceived investment in employee development and the intention to leave a job was fully explained by employee professional and organizational commitment, while for those without a government subsidy program, there were both direct and indirect effects of perceived investment in employee development on the intention to a leave job [39].

### 3.3. Turnover Intention

As shown in Table A1, six studies examined the relationship between turnover intention and professional development. Employees who had domestic training or overseas study outside of work had a turnover intention of 46%, while those who did not have any domestic training or overseas study had a turnover intention of 68% [25]. Several factors related to professional development, such as limited opportunities [24,35], inadequate continuing education [5], dissatisfaction with professional development [37], and low perceived investment in employee development [36] were associated with increased turnover intention. The effect of professional development opportunities on turnover intention differed by gender and profession [5,24]. A higher intention to leave the job was linked to inadequate professional development opportunities for men but not for women [24]. Similarly, doctors who had enough opportunities for continuing professional education had a lower intention to leave, but this was not the case for nurses [5].

Continuing professional education did not affect turnover intention for rural healthcare workers [5]. Moreover, the effect of satisfaction with professional development on turnover intention varied according to length of employment for nurses [37]. Satisfaction with professional development reduced turnover intention for nurses who had been employed for 13 to 18 months but not for those who had been employed for less than 12 months [37]. Training to improve skills or competences reduced turnover intention for nurses who had been employed for less than 6 months, while opportunities for professional development reduced turnover intention for nurses who had been employed for 7 to 24 months [35]. One study investigated the mechanisms underlying the link between perceived investment in employee development and turnover intention [36]. It found that job satisfaction and affective commitment fully mediated this link [36].

### 3.4. Return to Work, Job Change, Early Retirement, and Employment

Five studies were reviewed on different aspects of career transitions among workers (Table A1). A Danish study examined the effect of wage-subsidized job training on the duration of return to work and subsequent employment among sick-listed workers. The study found that the intervention shortened the time to return to non-subsidized work by three weeks but did not affect the stability of the subsequent employment [30]. Another study surveyed psychiatrists who moved or did not move to another area; it reported that professional support and development was a key factor in their decision to move to another area for 44% and 47% of them, respectively [26]. A third study investigated the motives for changing a job in the past five years among laboratory professionals from seven sub-Saharan African countries and revealed that the main reasons were lack of professional development or training (27.8%), lack of benefits (23.5%), relocation (22.6%), and poor working conditions (13.0%) [42]. A fourth study that analyzed the rate of early retirement among employees with different levels of domestic off-the-job training and/or overseas study showed that it was lower for those with some training or study (44%) than for those with none (63%) [25]. Lastly, the only study that investigated the relationship between professional development and employment status reported that participants who underwent training to enhance their professional skills had a higher probability of being employed than unemployed [34].

## 4. Discussion

The main finding of this systematic review is that there is a positive relationship between professional development or training and work participation. Employees who engage in skill development or training are more likely to stay in their current job than those who do not. However, the quality of the evidence is low, as 85% of the studies used a cross-sectional design, and more than a quarter of the studies did not adjust for confounding factors.

A review of the literature revealed that older nurses (over 45 years) have less access to continuing professional learning and development [43]. Only two cross-sectional studies examined the effect of age on the link between professional skill development and work participation [22,38]. One study found that continuing professional development was more important for retaining younger health professionals (mean age: 35.6 years) than older health professionals (mean age: 40.2 years) working with people with a disability in rural areas [22]. For younger health professionals, professional support, continuing professional development, and a high autonomy of practice were the main factors for staying in their current job, while for older health professionals, travel arrangements and a high autonomy of practice were the main factors. The other study reported that a lack of opportunities for professional development was a reason for leaving nursing care among both nurses younger than 45 years and those aged 45 years or older [38]. However, nurses younger than 45 years were more likely to leave nursing care due to the nursing workload and a low salary than older nurses aged ≥ 45 years [38]. Additionally, a qualitative study of 84 nurses over 50 years [44] that was not included in this review suggested that part-time work, flexible working hours, and continuing professional development could increase work participation. The relationship between continuing professional education and work retention among older workers has not been well studied. Future research using quasi-experimental and prospective cohort designs could help to determine if ongoing job training lowers the risk of leaving the workforce among ageing workers.

There is limited evidence on the mechanisms that explain how continuing professional education/training influences staying in a current job. Professional training is linked to higher organizational commitment [45]. Job satisfaction and affective commitment [40] or professional and organizational commitment [39] fully mediate the relationship between perceived investment in employee development and intention to quit a current job. Moreover, the association between perceived investment in employee development and employee turnover intention was fully mediated by job satisfaction and affective commitment [36]. However, one study did not find any indirect relationship between professional development and the intention to leave a profession through reducing burnout or increasing work engagement [6]. In addition to commitment and job satisfaction, continuing professional education enhances employees’ knowledge, skills, confidence, sense of coherence, work performance, and mental health, and it leads to changes in attitude, behavior, and practice [9,13]. These positive outcomes of continuing professional education can increase employee retention (Figure 2). Continuing professional education is related to higher career development and job satisfaction among workers [16,17,18,19,42], and to less work–family conflict, family complaints, and guilt regarding family [25]. Job satisfaction promotes employee retention [3,16], and job dissatisfaction is a common reason for quitting a job [23] or planning to retire early [46]. Mid-career physicians (41–60 years old) reported a lack of professional satisfaction as a more important factor influencing their retirement intention than physicians older than 60 years [46]. Previous studies have shown mixed results on the relationship between continuing professional training and work engagement. One study reported that workers who received professional training were more engaged in their work than workers without training [25], while another study found no link between continuing professional development and work engagement [6].

This review indicates that participating in professional training is linked to a lower risk of leaving current employment. However, continuing professional development opportunities are scarce, especially in remote and rural areas [5,7,26]. A major obstacle to accessing continuing professional development is a lack of financial support [14,47], and some workers have to pay for their own professional training [7]. Moreover, organizations reduce their budgets for professional learning and development during economic crises [48]; for example, the satisfaction rate with access to continuing professional development among rural Australian health care workers declined from 70% in 2005 to 35% in 2008 [49].

The current review has some limitations. Out of the 27 studies that explored this topic, most of them had a cross-sectional design (23 studies) and a small sample size. Only four studies had a cohort design, and a third of the studies did not adjust the observed associations for any confounding factor. In addition, few studies investigated the mechanisms and factors that influenced the effect of professional development/training on staying in a current job or in the labor market.

Future research should adopt more rigorous methods, such as quasi-experimental and longitudinal designs, to evaluate the impact of different types of training on the likelihood of exiting the labor force, especially among older workers. Future research should also identify which aspects of professional development and training are more effective or relevant for job retention than others (e.g., pedagogical aspects and personalization). Furthermore, in future research, it would be useful to differentiate between employees who leave their current job for a better one (qualitative employability) and those who leave for any other reason (quantitative employability) [50].

## 5. Conclusions

This review indicates that engaging in continuing professional training or development may help workers to retain their current employment. However, more high-quality studies, especially among older workers, are needed to examine the role of continuing job training in preventing labor-force exit.

## Figures and Tables

**Figure 1 healthcare-11-02900-f001:**
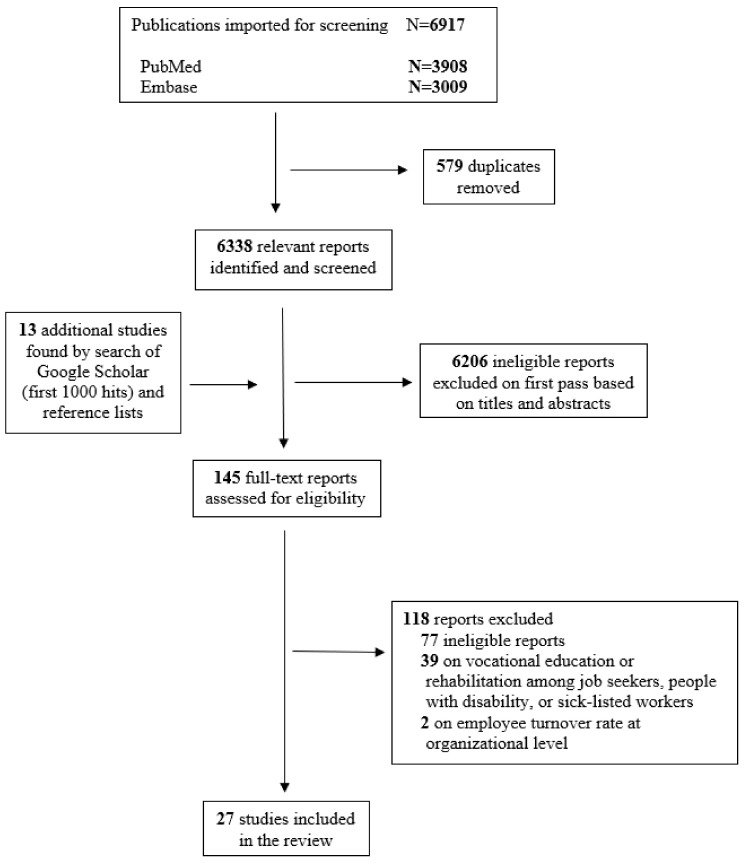
PRISMA flow diagram of the studies selection.

**Figure 2 healthcare-11-02900-f002:**
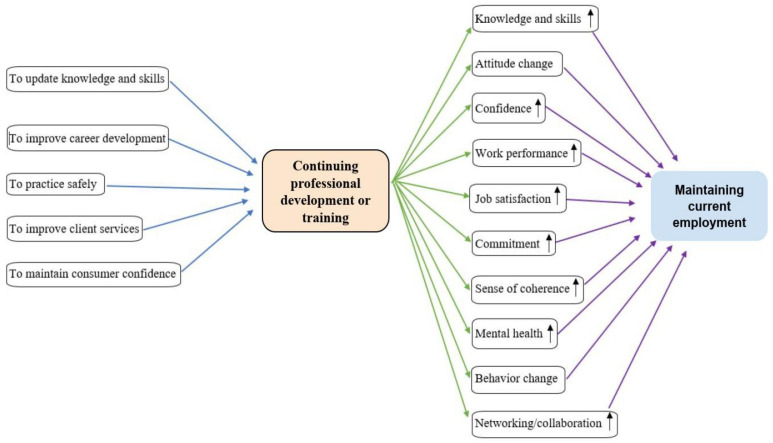
A conceptual diagram showing the potential mediators of the association between continuing professional development or training and maintaining current employment.

**Table 1 healthcare-11-02900-t001:** PubMed and Embase searches conducted on 2 January 2023.

Search	Query	No. of Items Found
PubMed		
#1	professional training[tiab]OR retraining[tiab] OR professional learning[tiab] OR relearning[tiab] OR reeducation[tiab] OR re-education[tiab] OR “education, professional, retraining”[Mesh] OR “vocational education”[Mesh] OR professional education[tiab] OR professional development[tiab] OR “education, continuing”[Mesh] OR continuing education[tiab] OR “interprofessional education”[Mesh] OR “inservice training”[Mesh] OR “staff development”[Mesh] OR job development[tiab] OR employee development[tiab] OR employees’ development[tiab] OR workplace learning[tiab] OR workplace training[tiab]	119,953
#2	work engagement[Mesh] OR “work engagement”[tiab] OR “employee participation”[tiab] OR “work participation”[tiab] OR “career participation”[tiab] OR “labor participation”[tiab] OR “labour participation”[tiab] OR “labor market participation”[tiab] OR “labour market participation”[tiab] OR employment[Mesh] OR unemployment[Mesh] OR return to work[Mesh] OR “return to work”[tiab] OR disability pension[tiab] OR disability retirement[tiab] OR retirement[Mesh] OR pensions[Mesh] OR early retirement[tiab] OR retired early[tiab] OR workforce recruitment[tiab] OR “workplace engagement”[tiab] OR workability[tiab] OR work ability[tiab] OR labor market exit[tiab] OR labour market exit[tiab] OR exit from employment[tiab] OR “personnel turnover”[Mesh]	127,609
#3	#1 AND #2	5162
#4	#3 Filters: Biography, case reports, comment, guideline, lecture, legal case, legislation, letter, editorial, news, newspaper article, portrait, published erratum, retracted publication, review, books and documents, case reports, dictionary, duplicate publication	665
#5	#3 NOT #4	4497
Final	#5 Filters: Humans	3908
Embase		
#1	‘interprofessional education’/exp OR ‘retraining’/exp OR ‘training’/mj OR ‘learning’/mj OR ‘skill retention’/exp OR ‘professional training’ OR ‘professional learning’ OR ‘relearning’ OR ‘reeducation’/exp OR ‘reeducation’ OR ‘re-education’ OR ‘vocational education’/exp OR ‘mentoring’/exp OR ‘lifelong learning’/exp OR ‘interdisciplinary education’/exp OR ‘in service training’/exp OR ‘continuing education’/exp OR ‘continuing education’ OR ‘adult education’/exp OR ‘refresher course’/exp OR ‘professional development’/exp OR ‘staff development’ OR ‘job development’ OR ‘employee development’ OR ‘employees development’ OR ‘workplace learning’ OR ‘workplace training’	228,705
#2	‘work engagement’/exp OR ‘work engagement’ OR ‘employee participation’ OR ‘work participation’ OR ‘career participation’ OR ‘labor participation’ OR ‘labour participation’ OR ‘labor market participation’ OR ‘labour market participation’ OR ‘employment’/exp OR ‘employment’ OR ‘unemployment’/exp OR ‘unemployment insurance’/exp OR ‘unemployment’ OR ‘return to work’/exp OR ‘return to work’ OR ‘disability pension’/exp OR ‘disability pension’ OR ‘disability retirement’ OR ‘retirement’/exp OR ‘early retirement’ OR ‘retired early’ OR ‘workforce recruitment’ OR ‘workplace engagement’ OR ‘workability’ OR ‘work ability’ OR ‘labor market exit’ OR ‘labour market exit’ OR ‘exit from employment’ OR ‘turnover rate’/exp OR ‘turnover rate’	251,313
#3	#1 AND #2	4289
#4	#3 AND (‘editorial’/it OR ‘letter’/it OR ‘note’/it OR ‘review’/it)	638
#4	#3 NOT #4	3651
Final	#5 AND ‘human’/de	3009

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
