# Peer review of "The Role of Continuing Professional Training or Development in Maintaining Current Employment: A Systematic Review"

_healthcare, 2023, doi:10.3390/healthcare11212900_

Round 1
Reviewer 1 Report
Comments and Suggestions for Authors
Thank you for the opportunity to review the interesting literature review. The topic is relevant and the efforts of the authors is very welcome and to be congratulated! However, there are some major concerns that I have that absolutely must be addressed in order to recommend acceptance of the manuscript.
1. I am a bit surprised that the included studies have such a large focus on health. It does seem very much like a study artifact, possibly due to the limitation to the databases chosen. I recommend, for one, using Google Scholar as an equivalent database (not limited to the first 1000 hits, as that was not done with the other databases either). In addition, there are established databases such as PsycArticles, PsyINFO, and Psyndex that should also be included.
2. There is a large amount of research on the effectiveness of training in different industries, so the statements in the introduction are not convincing. Reference should be made to the previous research on training in order to better identify the contribution of the study. In the discussion, the study should also be better integrated into the existing research on the topic of training. Some relevant studies:
Arthur, W., Bennett, W., Edens, P. S.& Bell, S. T. (2003). Effectiveness of training in organizations: A meta-analysis of design and evaluation features. Journal of Applied Psychology, 88(2), 234-245. https://doi.org/10.1037/0021-9010.88.2.234
Burke, M. J.& Day, R. R. (1986). A cumulative study of the effectiveness of managerial training. Journal of Applied Psychology, 71(2), 232-245. https://doi.org/10.1037/0021-9010.71.2.232
Collins, D. B.& Holton III, E. F. (2004). The effectiveness of managerial leadership development programs: A meta‐analysis of studies from 1982 to 2001. Human Resource Development Quarterly, 15(2), 217-248. https://doi.org/10.1002/hrdq.1099
Hughes, A. M., Gregory, M. E., Joseph, D. L., Sonesh, S. C., Marlow, S. L., Lacerenza, C. N., ... & Salas, E. (2016). Saving lives: A meta-analysis of team training in healthcare. Journal of Applied Psychology, 101(9), 1266-1304. https://doi.org/10.1037/apl0000120
Howard, M. C.& Gutworth, M. B. (2020). A meta-analysis of virtual reality training programs for social skill development. Computers & Education, 144, 103707. https://doi.org/10.1016/j.compedu.2019.103707
Kaplan, A. D., Cruit, J., Endsley, M., Beers, S. M., Sawyer, B. D.& Hancock, P. A. (2021). The effects of virtual reality, augmented reality, and mixed reality as training enhancement methods: A meta-analysis. Human Factors, 63(4), 706-726.
Lacerenza, C. N., Reyes, D. L., Marlow, S. L., Joseph, D. L.& Salas, E. (2017). Leadership training design, delivery, and implementation: A meta-analysis. Journal of Applied Psychology, 102(12), 1686-1718. https://doi.org/10.1037/apl0000241
McEwan, D., Ruissen, G. R., Eys, M. A., Zumbo, B. D.& Beauchamp, M. R. (2017). The effectiveness of teamwork training on teamwork behaviors and team performance: a systematic review and meta-analysis of controlled interventions. PloS One, 12(1), e0169604.
Powell, K. S., & Yalcin, S. (2010). Managerial training effectiveness: A meta‐analysis 1952‐2002. Personnel Review, 39(2), 227-241. https://doi.org/10.1108/00483481011017435
Salas, E., DiazGranados, D., Klein, C., Burke, C. S., Stagl, K. C., Goodwin, G. F.& Halpin, S. M. (2008). Does team training improve team performance? A meta-analysis. Human Factors, 50(6), 903-933.
Salas, E., Nichols, D. R.& Driskell, J. E. (2007). Testing three team training strategies in intact teams: A meta-analysis. Small Group Research, 38(4), 471-488.
3. In addition, the focus of the study is somewhat unclear: the title states "continuing professional training", i.e. a training focus, while the comments on page 2 refer to "continuing professional development". However, there is a fundamental difference here: while training can be equated with formal learning, development can be understood as all forms of learning, including, for example, informal and self regulated learning. There are already interesting studies (with a focus on similar outcomes) and even meta-analyses, e.g.:
Cerasoli, C. P., Alliger, G. M., Donsbach, J. S., Mathieu, J. E., Tannenbaum, S. I., & Orvis, K. A. (2018). Antecedents and outcomes of informal learning behaviors: A meta-analysis. Journal of Business and Psychology, 33, 203-230.
Decius, J., Knappstein, M., & Klug, K. (2023). Which Way of Learning Benefits Your Career? The Role of Different Forms of Work-related Learning for Different Types of Perceived Employability. European Journal of Work and Organizational Psychology. Advance Online Publication. https://doi.org/10.1080/1359432X.2023.2191846
Kortsch, T., Schulte, E. M., & Kauffeld, S. (2019). Learning@ work: informal learning strategies of German craft workers. European Journal of Training and Development, 43(5/6), 418-434.
Sitzmann, T.& Ely, K. (2011). A meta-analysis of self-regulated learning in work-related training and educational attainment: What we know and where we need to go. Psychological Bulletin, 137(3), 421–442. https://doi.org/10.1037/a0022777
4. It is also not convincingly explained why the study is limited to termination as an outcome and does not also include the increase in commitment (as a kind of antithesis to termination) and similar constructs such as internal market value/employability in order to get a more complete picture.
5. The paper also lacks a convincing theoretical basis linking "continuing professional development" to termination. One approach would be social exchange theory (Cropanzano & Mitchell, 2005), but this would need to be worked out better
6. Some minor points:
a. Varying font sizes in the text
b. Table 2 should be included in the appendix, because it is too complex for the text
Comments on the Quality of English Language
ok
Author Response
Thank you for the opportunity to review the interesting literature review. The topic is relevant and the efforts of the authors is very welcome and to be congratulated! However, there are some major concerns that I have that absolutely must be addressed in order to recommend acceptance of the manuscript.
Response: Thank you for your comments!
- I am a bit surprised that the included studies have such a large focus on health. It does seem very much like a study artifact, possibly due to the limitation to the databases chosen. I recommend, for one, using Google Scholar as an equivalent database (not limited to the first 1000 hits, as that was not done with the other databases either). In addition, there are established databases such as PsycArticles, PsyINFO, and Psyndex that should also be included.
Response. To conduct a systematic review, one should search at least one American database (such as PubMed and Web of Science) and at least one European database (such as Embase and Scopus). Google Scholar is not adequate for a systematic search and lacks advanced search features. We did not restrict our Google Scholar search to the first 1000 results. Google Scholar only allows screening the first 1000 results. PsycINFO is not pertinent to the current review. It is a database of psychological literature. Embase (41 million) and PubMed (34 million) cover the peer-reviewed articles indexed in PsycINFO (5 million). PsycArticles provides full texts of some of PsycINFO publications. PSYNDEX covers only psychological and psychology-related literature from German-speaking countries.
- There is a large amount of research on the effectiveness of training in different industries, so the statements in the introduction are not convincing. Reference should be made to the previous research on training in order to better identify the contribution of the study. In the discussion, the study should also be better integrated into the existing research on the topic of training. Some relevant studies:
Arthur, W., Bennett, W., Edens, P. S.& Bell, S. T. (2003). Effectiveness of training in organizations: A meta-analysis of design and evaluation features. Journal of Applied Psychology, 88(2), 234-245. https://doi.org/10.1037/0021-9010.88.2.234
Burke, M. J.& Day, R. R. (1986). A cumulative study of the effectiveness of managerial training. Journal of Applied Psychology, 71(2), 232-245. https://doi.org/10.1037/0021-9010.71.2.232
Collins, D. B.& Holton III, E. F. (2004). The effectiveness of managerial leadership development programs: A meta‐analysis of studies from 1982 to 2001. Human Resource Development Quarterly, 15(2), 217-248. https://doi.org/10.1002/hrdq.1099
Hughes, A. M., Gregory, M. E., Joseph, D. L., Sonesh, S. C., Marlow, S. L., Lacerenza, C. N., ... & Salas, E. (2016). Saving lives: A meta-analysis of team training in healthcare. Journal of Applied Psychology, 101(9), 1266-1304. https://doi.org/10.1037/apl0000120
Howard, M. C.& Gutworth, M. B. (2020). A meta-analysis of virtual reality training programs for social skill development. Computers & Education, 144, 103707. https://doi.org/10.1016/j.compedu.2019.103707
Kaplan, A. D., Cruit, J., Endsley, M., Beers, S. M., Sawyer, B. D.& Hancock, P. A. (2021). The effects of virtual reality, augmented reality, and mixed reality as training enhancement methods: A meta-analysis. Human Factors, 63(4), 706-726.
Lacerenza, C. N., Reyes, D. L., Marlow, S. L., Joseph, D. L.& Salas, E. (2017). Leadership training design, delivery, and implementation: A meta-analysis. Journal of Applied Psychology, 102(12), 1686-1718. https://doi.org/10.1037/apl0000241
McEwan, D., Ruissen, G. R., Eys, M. A., Zumbo, B. D.& Beauchamp, M. R. (2017). The effectiveness of teamwork training on teamwork behaviors and team performance: a systematic review and meta-analysis of controlled interventions. PloS One, 12(1), e0169604.
Powell, K. S., & Yalcin, S. (2010). Managerial training effectiveness: A meta‐analysis 1952‐2002. Personnel Review, 39(2), 227-241. https://doi.org/10.1108/00483481011017435
Salas, E., DiazGranados, D., Klein, C., Burke, C. S., Stagl, K. C., Goodwin, G. F.& Halpin, S. M. (2008). Does team training improve team performance? A meta-analysis. Human Factors, 50(6), 903-933.
Salas, E., Nichols, D. R.& Driskell, J. E. (2007). Testing three team training strategies in intact teams: A meta-analysis. Small Group Research, 38(4), 471-488.
Response. We screened these publications and did not find them sufficiently relevant to the aim of the current review. So, we could not use them in the introduction or discussion sections substantially. Arthur et al. 2003 investigated how training design and evaluation features influenced the outcomes of training in organizations. Burke & Day 1986 evaluated the impact of managerial training, Collins & Holton 2004 analyzed the literature on developing managerial leadership programs from 1982 to 2001, and Powell & Yalcin 2010 assessed the changes in the effectiveness of managerial training from 1952 to 2002. Hughes et al. 2016 conducted a study on healthcare team training to enhance teamwork. Howard et al. 2020 examined the effectiveness of virtual reality training programs for developing social skill. Kaplan et al. 2021 compared virtual, augmented, and mixed reality training with conventional training methods. Lacerenza et al. 2017 reviewed the literature on the impact of leadership training. Salas et al. 2007 explored different aspects of team training (cross-training, team coordination and adaptation training, and guided team self-correction training) on team performance. McEwan et al 2017 and Salas et al. 2008 evaluated the effects of team training interventions on improving team performance and teamwork.
- In addition, the focus of the study is somewhat unclear: the title states "continuing professional training", i.e. a training focus, while the comments on page 2 refer to "continuing professional development". However, there is a fundamental difference here: while training can be equated with formal learning, development can be understood as all forms of learning, including, for example, informal and self regulated learning. There are already interesting studies (with a focus on similar outcomes) and even meta-analyses, e.g.:
Cerasoli, C. P., Alliger, G. M., Donsbach, J. S., Mathieu, J. E., Tannenbaum, S. I., & Orvis, K. A. (2018). Antecedents and outcomes of informal learning behaviors: A meta-analysis. Journal of Business and Psychology, 33, 203-230.
Decius, J., Knappstein, M., & Klug, K. (2023). Which Way of Learning Benefits Your Career? The Role of Different Forms of Work-related Learning for Different Types of Perceived Employability. European Journal of Work and Organizational Psychology. Advance Online Publication. https://doi.org/10.1080/1359432X.2023.2191846
Kortsch, T., Schulte, E. M., & Kauffeld, S. (2019). Learning@ work: informal learning strategies of German craft workers. European Journal of Training and Development, 43(5/6), 418-434.
Sitzmann, T.& Ely, K. (2011). A meta-analysis of self-regulated learning in work-related training and educational attainment: What we know and where we need to go. Psychological Bulletin, 137(3), 421–442. https://doi.org/10.1037/a0022777
Response: We have revised the manuscript to make clear that we focus on both continuing professional training and professional development. We have used the study by Decius et al. 2023 in our discussion. The study by Cerasoli et al. 2018 on the antecedents of informal learning behaviors, the study by Kortsch et al. 2019 on informal learning strategies among craft workers, and the study by Sitzmann & Ely 2011 on self-regulation theory are not relevant for the current review.
- It is also not convincingly explained why the study is limited to termination as an outcome and does not also include the increase in commitment (as a kind of antithesis to termination) and similar constructs such as internal market value/employability in order to get a more complete picture.
Response: This is a systematic review that examines a specific research question. The review process involves setting inclusion and exclusion criteria beforehand. Commitment as an outcome cannot be considered as part of the decision to stay or leave a job.
- The paper also lacks a convincing theoretical basis linking "continuing professional development" to termination. One approach would be social exchange theory (Cropanzano & Mitchell, 2005), but this would need to be worked out better
Response: Social exchange theory does not fall within the scope of this review.
- Some minor points:
- Varying font sizes in the text
- Table 2 should be included in the appendix, because it is too complex for the text.
Response: We have corrected the font sizes and moved the table 2 to the appendix.
Reviewer 2 Report
Comments and Suggestions for Authors
Report for: The role of continuing professional training in maintaining current employment: A systematic review
Summary: The authors provide a literature review on the link between skill acquisition on-the-job and employment status. I think this is an interesting question. With technology being an ever-present component of work tasks, constant adjustments and adoption to new technology requires a willingness to pursue on-the-job training. This study sheds light on changes on employment status as one possible implication that might accompany the need for continuing skill acquisition.
Comments in more detail
1. Abstract:
a. replace expression “job education” with on-the-job training not just in the abstract but throughout the paper; this way you match the title and the literature that uses on-the-job training as an expression for skill acquisition on the job
b. line 19 on page 1: authors mention that majority of studies cover healthcare sector – do the authors have any intuition why that is the case and whether findings might (not) be applicable to other settings?
c. distinction in how employment status is measured (as mentioned in lines 19 through 21 on page 1) is neat – I would like to know whether similar heterogeneity/distinction exists in terms of how on-the-job training is measured as is it well known that on-the-job training can involve anything from very formal settings (e.g., seminars, workshops) to informal settings (e.g., informal meetings with a supervisor, observing others on the job, etc.)
d. the abstract would benefit if the findings could be positioned in the context of theory – how are we to understand the main takeaways that conclude the abstract? from personnel economics we know that type of capital (firm-specific vs. general) affects the gains from leaving the job after human capital is acquired through on-the-job training
2. Introduction:
a. the first paragraph in the introduction (bottom paragraph on page 1) focuses on retirement as an example of job exits but these are rare – why motivate the paper using an example that is probably one of the less common reasons people leave their jobs?
b. the way the topic is framed ignores the fact that workers can also get fired and that on-the-job skill acquisition can reduce the risk of firing – why do authors want to restrict their attention to the role training might play in explaining quits as training matters also for firings – expanding their discussion would make the paper more general
3. Results:
a. how are studies listed (how is the order determined) in Table 2? should the list be from the most recent to the oldest or are studies grouped based on measurement of y or x? I recommend grouping studies based on y or x given the focus of the review article
b. Table 2: sex distribution is not that interesting to warrant its own column; I would much prefer to know the years when people were surveyed as that may differ from the publication year
c. Table 2: it would be of interest if some assessment was made as to whether training provides general skills (that are transferable across firms) or firm-specific skills as that should be a key to explaining turnover
d. Table 2: rather than reporting the mean age of respondents I would be more interested in the mean of the key y and x: the mean of turnover measure and the mean of training recipients
e. sections 3.1 through 3.x: review the evidence on the basis of whether the skills acquired through training are general or firm-specific to improve our understanding of the findings
f. there is little discussion about the role labor market conditions play in turnover measures: existence of outside opportunities is key to someone switching jobs
Comments on the Quality of English LanguagePlease see above for comments for the authors.
Author Response
Report for: The role of continuing professional training in maintaining current employment: A systematic review
Summary: The authors provide a literature review on the link between skill acquisition on-the-job and employment status. I think this is an interesting question. With technology being an ever-present component of work tasks, constant adjustments and adoption to new technology requires a willingness to pursue on-the-job training. This study sheds light on changes on employment status as one possible implication that might accompany the need for continuing skill acquisition.
Response: Thank you for your comments!
Comments in more detail
- Abstract:
- replace expression “job education” with on-the-job training not just in the abstract but throughout the paper; this way you match the title and the literature that uses on-the-job training as an expression for skill acquisition on the job.
Response. Continuing professional education cannot be substituted by on-the-job training. The association between on-the-job training and job retention, leaving a job, or turnover was only investigated by some of the studies included in the review.
- line 19 on page 1: authors mention that majority of studies cover healthcare sector – do the authors have any intuition why that is the case and whether findings might (not) be applicable to other settings?
Response. The majority of the studies in this review (89%) focused on health care workers. Therefore, the results of this review may be applicable to other professions as well. However, this review cannot make any claims about other professions.
- distinction in how employment status is measured (as mentioned in lines 19 through 21 on page 1) is neat – I would like to know whether similar heterogeneity/distinction exists in terms of how on-the-job training is measured as is it well known that on-the-job training can involve anything from very formal settings (e.g., seminars, workshops) to informal settings (e.g., informal meetings with a supervisor, observing others on the job, etc.).
Response. We have extracted all information reported in the included studies regarding continuing professional education/development and summarized the data in Table 2. We did not find any other relevant information to supplement Table 2.
- the abstract would benefit if the findings could be positioned in the context of theory – how are we to understand the main takeaways that conclude the abstract? from personnel economics we know that type of capital (firm-specific vs. general) affects the gains from leaving the job after human capital is acquired through on-the-job training.
Response. The review question we examined was specific, and any general conclusion drawn from the review is not supported by the evidence. The data showed a link between ongoing job training and keeping the same job, and this link was fully accounted for by job satisfaction and commitment. Continuing job training had an indirect effect on job retention.
- Introduction:
- the first paragraph in the introduction (bottom paragraph on page 1) focuses on retirement as an example of job exits but these are rare – why motivate the paper using an example that is probably one of the less common reasons people leave their jobs?
Response. We have revised the sentence to omit the examples. One of the factors for quitting a job was “early retirement”. The influence of continuing education on early retirement was investigated in one of the included studies.
- the way the topic is framed ignores the fact that workers can also get fired and that on-the-job skill acquisition can reduce the risk of firing – why do authors want to restrict their attention to the role training might play in explaining quits as training matters also for firings – expanding their discussion would make the paper more general.
Response. No research has examined how continuing education or professional development affects firing. A PubMed query does not yield any results on the link between continuing education or professional development and job firing or job quitting.
- Results:
- how are studies listed (how is the order determined) in Table 2? should the list be from the most recent to the oldest or are studies grouped based on measurement of y or x? I recommend grouping studies based on y or x given the focus of the review article
Response. We have organized the studies based on the study design (cohort studies first and then cross-sectional studies) because cohort studies have more validity than cross-sectional studies. Then we have arranged the studies in reverse chronological order from newest to oldest studies. We have added this information to the title of Table 2.
- Table 2: sex distribution is not that interesting to warrant its own column; I would much prefer to know the years when people were surveyed as that may differ from the publication year.
Response. Our analysis focused on how continuing professional education or development relates to job retention, job departure, or turnover rate. We did not examine the frequency or proportion of continuing professional education or development, or the frequency or proportion of staying or leaving a job. The year of data collection does not affect the relationship we observed. Furthermore, not all the studies we included reported the year of data collection. Some of the studies reported that the relationship varies by gender.
- Table 2: it would be of interest if some assessment was made as to whether training provides general skills (that are transferable across firms) or firm-specific skills as that should be a key to explaining turnover.
Response. The studies included in this review did not report such information.
- Table 2: rather than reporting the mean age of respondents I would be more interested in the mean of the key y and x: the mean of turnover measure and the mean of training recipients.
Response. We have extracted all information reported in the included studies regarding continuing professional education/development and staying/leaving a job or turnover and summarized the data from the studies in Table 2. We did not find any other relevant information to supplement Table 2.
- sections 3.1 through 3.x: review the evidence on the basis of whether the skills acquired through training are general or firm-specific to improve our understanding of the findings.
Response. The studies included in this review did not report such information.
- there is little discussion about the role labor market conditions play in turnover measures: existence of outside opportunities is key to someone switching jobs.
Response. We have discussed this issue at the end of our discussion section.
Reviewer 3 Report
Comments and Suggestions for Authors
The authors present a paper entitled “The role of continuing professional training in maintaining current employment: a systematic review”.
· Personally, I think the tables in the work take up an excessive amount of space.
· The following sentence: "Participants who received professional skill development training were more than six times more likely to be employed than unemployed [34]" does not seem relevant to me in this context.
· Although the paper focuses on the effects of training in the medical field, in the section on job retention none of the work presented refers to that aera.
· In general, it seems to me that the formatting needs to be revised.
· Please explain more in details the content of the following statement: “The effect of professional development opportunities on turnover intention was significant for men but not for women [24], and the effect of continuing professional education on turnover intention was significant for doctors but not for nurses [5].”
· I find that the diagram depicted in Figure 2 does not exhibit precise alignment with the textual content, nor does it possess particularly appealing aesthetics.
· I believe one of the main critical issues in the work is the difficulty in understanding and defining what is meant by development (Training): is a 4-day training the same as a 20-day training? There seems to me to be a lot of heterogeneity in this regard. We should also consider that sometimes, trainings are on things that are of no interest, had these aspects been measured and how can they play a role?
Since continuing professional training programs vary widely in content, quality, and delivery methods, assessing the effectiveness of all programs as a whole may not accurately represent the specific program(s) in question.
Comments on the Quality of English LanguageThe overall quality of the English is sufficient, with only a few minor issues that need checking.
Author Response
The authors present a paper entitled “The role of continuing professional training in maintaining current employment: a systematic review”.
Personally, I think the tables in the work take up an excessive amount of space.
Response: Thank you for your comments!
We have moved the table 2 to the appendix. There is no word limit for submissions to open access journals and most of the readers do not pay attention to supplementary results. This way (adding to the appendix), we can present our main findings more clearly and concisely.
The following sentence: "Participants who received professional skill development training were more than six times more likely to be employed than unemployed [34]" does not seem relevant to me in this context.
Response: We have moved this study to the last portion of the results and presented this investigation in the section on “employment”.
Although the paper focuses on the effects of training in the medical field, in the section on job retention none of the work presented refers to that aera.
In general, it seems to me that the formatting needs to be revised.
Response: This review aimed to investigate how continuing professional education or development influences the decision to stay or leave a job. Turnover refers to quitting or switching jobs. It was not justified to exclude these studies from the review.
Please explain more in details the content of the following statement: “The effect of professional development opportunities on turnover intention was significant for men but not for women [24], and the effect of continuing professional education on turnover intention was significant for doctors but not for nurses [5].”
Response: We have revised these results on page 6 and replaced with the following statements:
A higher intention to leave the job was linked to inadequate professional development opportunities for men, but not for women [24]. Similarly, doctors who had enough opportunities for continuing professional education had a lower intention to leave, but this was not the case for nurses [5].
I find that the diagram depicted in Figure 2 does not exhibit precise alignment with the textual content, nor does it possess particularly appealing aesthetics.
Response: We have adjusted the diagram's content to be more consistent with the text. We also enhanced the aesthetic quality of the diagram.
I believe one of the main critical issues in the work is the difficulty in understanding and defining what is meant by development (Training): is a 4-day training the same as a 20-day training? There seems to me to be a lot of heterogeneity in this regard. We should also consider that sometimes, trainings are on things that are of no interest, had these aspects been measured and how can they play a role?
Response: We did not conduct a meta-analysis and did not develop a guideline for continuing professional education or development to concern about heterogeneity. We only reported the presence or absence of an association.
Since continuing professional training programs vary widely in content, quality, and delivery methods, assessing the effectiveness of all programs as a whole may not accurately represent the specific program(s) in question.
Response: This is a qualitative systematic review that examined the presence or absence of an association between continuing education and job retention. We did not perform any meta-analysis to quantitatively pool the results from the included studies. All the studies we selected compared groups with or without continuing education or professional development or compared different training frequencies. The results suggest that continuing professional education or development has a positive effect on staying in the current job, but this effect is mediated by increased job satisfaction and commitment. The different content, quality, delivery method, and amount of continuing education were positively associated with job retention.
Round 2
Reviewer 1 Report
Comments and Suggestions for Authors
Thank you very much for the answers. In general, I would have liked the authors to make more of an effort to adequately address my comments. After all, a review is a feedback offer that researchers create in their spare time without compensation. For the more general criticisms, I find relatively little in the text and only marginal changes.
Comment 1: To conduct a systematic review, one should search at least one American database (such as PubMed and Web of Science) and at least one European database (such as Embase and Scopus). Google Scholar is not adequate for a systematic search and lacks advanced search features. We did not restrict our Google Scholar search to the first 1000 results. Google Scholar only allows screening the first 1000 results. PsycINFO is not pertinent to the current review. It is a database of psychological literature. Embase (41 million) and PubMed (34 million) cover the peer-reviewed articles indexed in PsycINFO (5 million). PsycArticles provides full texts of some of PsycINFO publications. PSYNDEX covers only psychological and psychology-related literature from German-speaking countries.
Response: Thank you for the clarification. What still does not convince me is that one of the findings is: "Out of the 27 studies, 22 involved health care workers as the target population, while 2 focused on faculty members of health or medical sciences." At the same time, the group of "health care workers" is already highlighted in the introduction (“Continuing professional development is more common among health care workers [9], while its benefits for other occupations are less explored. Health care workers participate in continuing professional education and training to develop their careers, stay updated and improve the quality of patient care”). However, judging from the search terms, the search is obviously not limited to this group. Thus, I am still not convinced whether the result is not a methodological artifact. If the question really refers to the learning and development of health care workers and job retention - which I would find interesting and relevant - this should of course be reflected in the search terms. So how can you place the findings in the research to date?
The proposed meta-analyses, even if they have a slightly different focus, did not find such a systematic pattern of findings with a focus on health care workers. How is it then that they come to such a conclusion? That still needs a better explanation. To be clear: I don't think their result is wrong, but it is so surprising to me that it needs to be further validated and then embedded appropriately in previous research. In doing so, I would also like them to reconsider how they fit their review into previous training research. All
Concerning your point “We did not restrict our Google Scholar search to the first 1000 results. Google Scholar only allows screening the first 1000 results.” à Please also explain this in the manuscript to avoid such misunderstandings. And there are also strategies to cope with the restriction (e. g., dividing the search between different date in the 'Return articles dated between' of advanced search area of GS).
Comment: We screened these publications and did not find them sufficiently relevant to the aim of the current review. So, we could not use them in the introduction or discussion sections substantially. Arthur et al. 2003 investigated how training design and evaluation features influenced the outcomes of training in organizations. Burke & Day 1986 evaluated the impact of managerial training, Collins & Holton 2004 analyzed the literature on developing managerial leadership programs from 1982 to 2001, and Powell & Yalcin 2010 assessed the changes in the effectiveness of managerial training from 1952 to 2002. Hughes et al. 2016 conducted a study on healthcare team training to enhance teamwork. Howard et al. 2020 examined the effectiveness of virtual reality training programs for developing social skill. Kaplan et al. 2021 compared virtual, augmented, and mixed reality training with conventional training methods. Lacerenza et al. 2017 reviewed the literature on the impact of leadership training. Salas et al. 2007 explored different aspects of team training (cross-training, team coordination and adaptation training, and guided team self-correction training) on team performance. McEwan et al 2017 and Salas et al. 2008 evaluated the effects of team training interventions on improving team performance and teamwork.
Response: Please see my other points.
Comment: We have revised the manuscript to make clear that we focus on both continuing professional training and professional development. We have used the study by Decius et al. 2023 in our discussion. The study by Cerasoli et al. 2018 on the antecedents of informal learning behaviors, the study by Kortsch et al. 2019 on informal learning strategies among craft workers, and the study by Sitzmann & Ely 2011 on self-regulation theory are not relevant for the current review.
Response: Thank you for the answer. However, in line 64, p. 2, it is still imprecisely written "the effect of job education or training" when deriving the study objective. This should be clarified again. In addition, I also ask again my points from the next answers to consider.
Comment: This is a systematic review that examines a specific research question. The review process involves setting inclusion and exclusion criteria beforehand. Commitment as an outcome cannot be considered as part of the decision to stay or leave a job.
Response: I understand the point, but it is more of a methodological argument. My criticism is aimed at the theoretical justification of the selection of these criteria. Here please also note my answer to the next point of criticism. As long as there is no good theoretical justification for why one is interested in something (and excludes others by the theoretical approach), such a selection of criteria always seems somewhat arbitrary. In this respect, my criticism is in no way dispelled by the answer; I consider the aspects I mentioned to be theoretically relevant, but I am happy to hear theoretical arguments that justify their focus in a comprehensible way.
Comment: Social exchange theory does not fall within the scope of this review.
Response: My critique does not refer to including social exchange theory in the search. Rather, I find it unconvincingly presented in the introduction why the effect of "job education or training on staying or leaving the current employment" is examined, i.e. why "continuing professional training or development" is relevant for "staying or leaving." So I am missing a theoretical justification, there the social exchange theory would be a suggestion, but maybe you have a different theoretical approach there. In any case, such a theoretical justification should be added to the relevance arguments that have already been given.
Comment: We have corrected the font sizes and moved the table 2 to the appendix.
Response: Thank you.
Author Response
Thank you very much for the answers. In general, I would have liked the authors to make more of an effort to adequately address my comments. After all, a review is a feedback offer that researchers create in their spare time without compensation. For the more general criticisms, I find relatively little in the text and only marginal changes.
Response: We appreciate your feedback on our report, and we welcome any suggestions that can help us enhance its accuracy and clarity. However, we do not find it relevant or helpful to refer to meta-analyses that are beyond the scope of our review. Our report is focused on a specific topic, and we have followed rigorous criteria to select the studies that are most pertinent and reliable. Therefore, we kindly ask you to respect the scope and methodology of our report and refrain from introducing sources that are not directly related to it.
Thank you for the clarification. What still does not convince me is that one of the findings is: "Out of the 27 studies, 22 involved health care workers as the target population, while 2 focused on faculty members of health or medical sciences." At the same time, the group of "health care workers" is already highlighted in the introduction (“Continuing professional development is more common among health care workers [9], while its benefits for other occupations are less explored. Health care workers participate in continuing professional education and training to develop their careers, stay updated and improve the quality of patient care”). However, judging from the search terms, the search is obviously not limited to this group. Thus, I am still not convinced whether the result is not a methodological artifact. If the question really refers to the learning and development of health care workers and job retention - which I would find interesting and relevant - this should of course be reflected in the search terms. So how can you place the findings in the research to date?
Response: The importance of continuing professional training for healthcare professionals has been widely researched, while other professions have received less attention. This is understandable, as healthcare professionals are required to update their knowledge and skills in many countries. Healthcare is a unique field, where the lives of people are at stake. Therefore, professional training is more prevalent and essential among healthcare workers. We used the PICOTS framework to guide our searches and did not restrict our search criteria by population, study design, or setting. You can verify the comprehensiveness of our search by conducting your own search in one database.
The proposed meta-analyses, even if they have a slightly different focus, did not find such a systematic pattern of findings with a focus on health care workers. How is it then that they come to such a conclusion? That still needs a better explanation. To be clear: I don't think their result is wrong, but it is so surprising to me that it needs to be further validated and then embedded appropriately in previous research. In doing so, I would also like them to reconsider how they fit their review into previous training research.
Response: The main focus of meta-analytic reviews you proposed is not about individual continuing professional training. Rather, you have selected studies that examined different aspects of organizational learning, such as managerial leadership, virtual reality training, team training, and team performance. These are distinct from individual professional job training in terms of their goals, methods, outcomes, and contexts. Therefore, those meta-analytic reviews do not address the research question that we have explored in this review.
Concerning your point “We did not restrict our Google Scholar search to the first 1000 results. Google Scholar only allows screening the first 1000 results.” à Please also explain this in the manuscript to avoid such misunderstandings. And there are also strategies to cope with the restriction (e. g., dividing the search between different date in the 'Return articles dated between' of advanced search area of GS).
Response: We have added the following statement to the results section.
“Google Scholar only allows screening the first 1000 results.”
In line 64, p. 2, it is still imprecisely written "the effect of job education or training" when deriving the study objective. This should be clarified again. In addition, I also ask again my points from the next answers to consider.
Response: We have added “professional development” to our statement.
I understand the point, but it is more of a methodological argument. My criticism is aimed at the theoretical justification of the selection of these criteria. Here please also note my answer to the next point of criticism. As long as there is no good theoretical justification for why one is interested in something (and excludes others by the theoretical approach), such a selection of criteria always seems somewhat arbitrary. In this respect, my criticism is in no way dispelled by the answer; I consider the aspects I mentioned to be theoretically relevant, but I am happy to hear theoretical arguments that justify their focus in a comprehensible way.
Response: Continuing professional training is crucial for healthcare workers. It enables them to keep up with the latest developments in their fields and to provide high-quality care to their patients. However, there is a lack of theoretical understanding of how continuing professional training affects the retention of healthcare workers in their jobs. This gap in knowledge motivated us to develop a conceptual model (Figure 2) that can explain the observed association between continuing professional training and job retention. The model aims to identify the factors that influence the decision of healthcare workers to stay or leave their jobs, and how continuing professional training can affect those factors.
My critique does not refer to including social exchange theory in the search. Rather, I find it unconvincingly presented in the introduction why the effect of "job education or training on staying or leaving the current employment" is examined, i.e. why "continuing professional training or development" is relevant for "staying or leaving." So I am missing a theoretical justification, there the social exchange theory would be a suggestion, but maybe you have a different theoretical approach there. In any case, such a theoretical justification should be added to the relevance arguments that have already been given.
Response: There is the lack of a clear theoretical framework to present it in the introduction. The relation between continuing professional training and job retention cannot be explained by the social exchange theory, which assumes that individuals engage in social interactions based on a rational calculation of costs and benefits. This theory does not capture the complexity and ethical implications of healthcare work, where the well-being and safety of patients are paramount. As we will explain in the discussion section, we have addressed this gap by drawing a conceptual diagram. Our main argument is that continuing professional training can be seen as a form of investment that enhances the job satisfaction and commitment of workers, and thus reduces their intention to leave their jobs or the profession.